# Breaking Physical and Linguistic Borders: Multilingual Federated Prompt Tuning for Low-Resource Languages

**Wanru Zhao**[1][*]**, Yihong Chen**[2]**, Royson Lee**[1,3]**, Xinchi Qiu**[1]
**Yan Gao**[1,4] **Hongxiang Fan**[1,3]**, Nicholas D. Lane** [1,4]

[1] University of Cambridge, [2] University College London,
[3] Samsung AI Center, Cambridge, [4] Flower Labs

## Abstract

Pre-trained large language models (LLMs) have emerged as a cornerstone in modern natural language processing, with their utility expanding to various applications and languages. However, the fine-tuning of multilingual LLMs, particularly for low-resource languages, is fraught with challenges steming from data-sharing restrictions (the physical border) and from the inherent linguistic differences (the linguistic border). These barriers hinder users of various languages, especially those in low-resource regions, from fully benefiting from the advantages of LLMs.

To address these challenges, we propose the Federated Prompt Tuning Paradigm for multilingual scenarios, which utilizes parameter-efficient fine-tuning while adhering to data sharing restrictions. We have designed a comprehensive set of experiments and analyzed them using a novel notion of language distance to underscore the strengths of this paradigm: Even under computational constraints, our method not only bolsters data efficiency but also facilitates mutual enhancements across languages, particularly benefiting low-resource ones. Compared to traditional local cross-lingual transfer tuning methods, our approach achieves 6.9% higher accuracy, reduces the training parameters by over 99%, and demonstrates better stability and generalization. Such findings underscore the potential of our approach to promote social equality and champion linguistic diversity, so that no language will be left behind. Our code is released at https://github.com/Ryan0v0/multilingual_borders.

## 1 Introduction

Large language models (LLMs) have been driving the recent progress in natural language processing (Brown et al., 2020; Chowdhery et al., 2022; Anil et al., 2023; Touvron et al., 2023a;b). These large models, built on extensive corpora, offer valuable insights and impressive results across a range of applications. In the meantime, in order to provide universally accessible knowledge with LLMs, extending the LLMs to multiple languages has become a particularly relevant research target (Conneau & Lample, 2019; Conneau et al., 2020; Artetxe et al., 2020)

Fine-tuning and deploying multilingual LLMs for practical downstream tasks present a unique set of challenges, distinct from those encountered with monolingual models. A primary concern is the geographical distribution of data across different languages, often stored in separate physical locations, making the sharing of data across regions difficult or, in some cases, prohibited due to legal constraints. For example, regulations such as the General Data Protection Regulation (GDPR) impose significant limitations on cross-region data sharing (Lim et al., 2020). Additionally, the linguistic diversity across regions, such as the differences between Sino-Tibetan and Indo-European languages, introduces a Non-Independent and Identically Distributed (non-IID) challenge in learning a unified multilingual model. This situation accentuates privacy concerns and highlights the need for effective privacy-preserving techniques when using multilingual LLMs. To this end, some recent works attempt to address privacy-constrained fine-tuning for multilingual tasks and explore how

---
[*]Corresponding to: Wanru Zhao (wz341@cam.ac.uk)

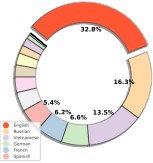 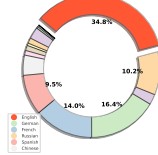 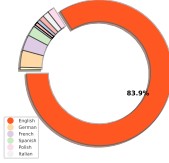

(a) XLM-R (Conneau et al., 2020)   (b) mBERT (Pires et al., 2019a)   (c) PaLM (Chowdhery et al., 2022)

Figure 1: Linguistic coverage of different large language models.

different languages impact the federated process (Weller et al., 2022). However, these works primarily target high-resources languages, leaving the low-resource languages under-explored.

Addressing low-resource languages is essential to promoting technological fairness and protecting linguistic diversity. Unlike their high-resource counterparts, low-resource languages pose intriguing research challenges: i) **Limited computational resources.** Regions of low-resource languages are often economically developing areas, with little access to enough computational resources required to either train language models from scratch or fully fine-tune pre-trained large language models (Mager et al., 2021; Adebara & Abdul-Mageed, 2022). ii) **Limited data in the target language.** Due to a small speaking population or the spoken nature of the language, data is often scarce (Adelani et al., 2021; Muhammad et al., 2022; Ebrahimi et al., 2022). As depicted in Figure 1, the pre-training data for LLMs is predominantly in English, with little coverage of low-resource languages. Under such circumstances, the performance of low-resource languages is often unsatisfactory during fine-tuning because of under-representation. iii) **Memorization risk.** Recent studies found that as pre-trained models scale up, their ability to memorize training data increases (Tirumala et al., 2022), which implies that, when fine-tuning these models with limited data, the risk of overfitting and potential privacy issues arises.

Recognizing the challenges posed by low-resource languages, Federated Learning (FL) has emerged as a promising training paradigm that addresses many of these concerns. In FL, model training occurs across multiple decentralized devices or servers, with the critical distinction that data remains localized McMahan et al. (2017); Beutel et al. (2020); Mathur et al. (2021). This approach is particularly well-suited to multilingual settings, where leveraging data from diverse linguistic backgrounds is essential without compromising data privacy. The geographical distribution and inherent linguistic diversity of devices in FL mean that the data on each node are likely to exhibit a non-IID (Non-Independent and Identically Distributed) distribution. This inherent characteristic of FL not only aligns with the privacy-preserving needs but also naturally accommodates the varied and complex linguistic features of low-resource languages, making it an ideal methodology for enhancing the inclusivity and effectiveness of language technologies. Furthermore, the efficiency of computational and communication processes Qiu et al. (2023; 2022) is paramount in such scenarios, underscoring the need for FL approaches that are not only privacy-centric but also resource-efficient, ensuring broad applicability and sustainability.

In this paper, in order to break the geographic border and the linguistic border between different language speaking countries, we propose a new paradigm grounded in FL, Multilingual Federated Prompt Tuning, focusing on parameter-efficient fine-tuning for multilingual tasks across various regions or devices. Specifically, by having each country fine-tune the model locally and then pass the updated parameters to a server for aggregation, we obtain a global model, which leverages collective knowledge and exposing models to a wider range of linguistic patterns. Considering the different linguistic patterns in various countries, our prompt encoders help generalize and adapt to the languages of different countries. We demonstrate the effectiveness of our paradigm on standard NLP tasks. The performance of our paradigm achieves 6.9% accuracy improvement while navigating privacy regulations that restrict cross-country data sharing. Compared with local monolingual fine-tuning, our paradigm reduces computational and communication cost by more than 99%. Our approach paves the way for fine-tuning multilingual large language models on resource-constraint devices across various regions, and holds the potential to promote social equality, privacy, and linguistic diversity in the research community. Our contributions are as follows:

- We demonstrate that federated prompt tuning can serve as a new paradigm for addressing the linguistic and physical challenges of multilingual fine-tuning across regions or devices.

- Compared to traditional local monolingual fine-tuning paradigm, federated prompt tuning is not only data and parameter efficient, suitable for situations with limited computational power, but also shows better generalization and stability, performing well in downstream tasks of low-resource languages with huge *language distance* from the *pre-trained language*.

- Federated prompt tuning also helps to alleviating data privacy leakage by reducing both the data transmission amount and data memorization, which opens up new avenues for exploration and has the potential to inform future research in this area.

## 2 RELATED WORK

**Multilingual Language Models.** Multilingual Pre-trained Language Models (PLMs) such as mBERT (Pires et al., 2019a), XLM-R (Conneau et al., 2020), and SeamlessM4T (Loic Barrault, 2023) have emerged as a viable option for bringing the power of pre-training to a large number of languages (Doddapaneni et al., 2021). Many studies analyzed mBERT's and XLM-R's capabilities and limitations, finding that the multilingual models work surprisingly well for cross-lingual tasks, despite the fact that they do not rely on direct cross-lingual supervision (e.g., parallel or comparable data, and translation dictionaries (Pires et al., 2019b; Wu & Dredze, 2019; Artetxe et al., 2020)

However, these multilingual PLMs are not without limitations. Particularly, Conneau et al. (2020) observed the *curse of multilinguality* phenomenon: given a fixed model capacity, adding more languages does not necessarily improve the multilingual performance but can deteriorate the performance after a certain point, especially for underrepresented languages (Wu & Dredze, 2020; Hu et al., 2020; Lauscher et al., 2020) Prior work tried to address this issue by increasing the model capacity (Artetxe et al., 2020; Pfeiffer et al., 2020; Chau et al., 2020) or through additional training for particular language pairs (Pfeiffer et al., 2020; Ponti et al., 2020) or by clustering and merging the vocabularies of similar languages, before defining a joint vocabulary across all languages (Chung et al., 2020). Despite these efforts, the multilingual PLMs still struggle with balancing their capacity across many languages in an sample-efficient and parameter-efficient way (Ansell et al., 2022; Marchisio et al., 2022; Chen et al., 2023).

**Prompt Learning and Parameter-Efficient fine-tuning.** The size of pre-trained language models has been increasing significantly (Brown et al., 2020), presenting challenges to traditional task transfer based on full-parameter fine-tuning. Recent research has shifted its attention to Parameter-Efficient fine-tuning techniques, such as prompt tuning (Lester et al., 2021; Li & Liang, 2021; Liu et al., 2021b), adapters (Houlsby et al., 2019a), as well as combined approaches including LoRA (Hu et al., 2021) and BitFit (Ben Zaken et al., 2022). These methods utilize a minimal number of tuning parameters, yet they offer transfer performance that is comparable with traditional fine-tuning.

Prompt learning involves training a small set of prompt tokens while keeping the original pre-trained model parameters frozen, thus allowing for model personalization with minimal parameter updates. (Liu et al., 2021a). This paradigm shows promise in effectively leveraging large pre-trained models in a data-efficient manner by reducing the need for extensive labeled datasets (Schick & Schütze, 2022). Additionally, prompt learning has exhibited a remarkable ability to generalize across a variety of tasks, suggesting a step towards more flexible and adaptable machine learning systems (Shin et al., 2020). Another most widely used Parameter-Efficient fine-tuning technique is LoRA, or Low-Rank Adaptation, which involves freezing the pre-trained model weights and injecting trainable rank decomposition matrices into each layer of the Transformer architecture, thereby achieving fine-tuning without incurring any additional inference latency (Hu et al., 2021).

**Federated Learning.** Federated Learning has garnered significant attention in the academic realm. It bypasses the conventional model training process by sharing models instead of raw data. With Federated Averaging (FedAvg) (McMahan et al., 2017), participating clients train models using their respective private datasets locally, and the updated model parameters are aggregated. This preserves the privacy of the underlying data while collectively benefiting from the knowledge gained during the training process (Konečný et al., 2016). Despite abundant research made on problems at hospitals, legal firms, and financial institutions, extending language models for multilingual usages effectively and efficiently, especially for low-resource languages remains under-explored.

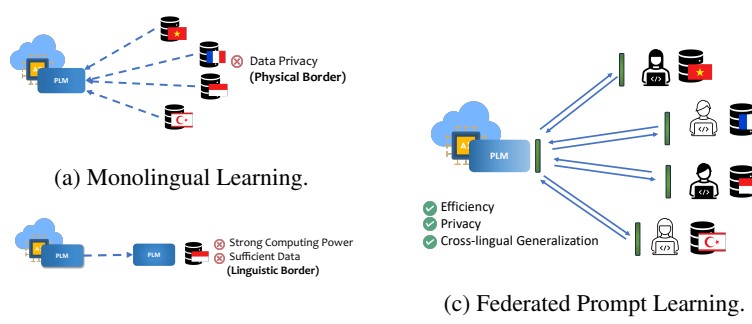

(a) Monolingual Learning.

(b) Centralized Learning.

(c) Federated Prompt Learning.

Figure 2: Comparison of three different learning paradigms for multilingual tasks.

In the general NLP domain, FL has been instrumental in tasks such as language modeling, sentiment analysis, and machine translation, showcasing its potential to revolutionize the way models are trained and deployed (Banabilah et al., 2022). Lin et al. (2022) introduces a benchmarking framework for evaluating various FL methods across NLP tasks, providing a universal interface between Transformer-based models and FL methods. Wang et al. (2022)is a federated approach designed for multilingual Natural Language Understanding (NLU) that integrates knowledge from multiple data sources through federated learning techniques to enhance the efficacy and accuracy of multilingual text processing. However, considerations regarding computational and communication efficiency in resource-constrained environments have not been adequately addressed.

## 3 A NEW PARADIGM FOR MULTILINGUALITY: FEDERATED PROMPT TUNING

In our federated learning setup, we have $K$ clients. Each client $k$ has a private dataset, either monolingual or multilingual, defined as: $\mathcal{D}_k = \{(x_{k,i}, y_{k,i}) \mid i = 1, \ldots, n_k\}$, where $x_{k,i}$ denotes the textual content, and $y_{k,i}$ is its corresponding label. The server sets up and maintains a global prompt encoder. We denote the parameters of the global prompt encoder as $h_g$, and each client $k$ has its own prompt encoder, denoted its parameters by $h_k$, tuned based on its dataset.

### 3.1 VIRTUAL PROMPT ENCODER

Instead of selecting discrete text prompts in a manual or automated fashion, in our Multilingual Federated Prompt Tuning paradigm, we utilize virtual prompt embeddings that can be optimized via gradient descent. Specifically, each prompt encoder, whether global or local, takes a series of virtual tokens, which are updated during tuning to better aid the model.

Figure 3 shows how our prompt tuning works on both clients and server. Specifically, on each client $k$, a textual prompt tailored for a specific task and input text $x$ are passed to the model. Then task specific virtual tokens are retrieved based on the textual prompt.

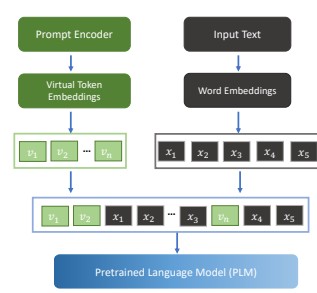

Figure 3: The pipeline of prompt tuning.

The primary objective of each prompt encoder is to generate an effective prompt embedding $v_0, v_1, v_2, \ldots$ for each client based on task specific virtual tokens, to guide the PLM in producing the desired outputs. With the input text tokenized, the discrete word token embeddings $x_1, x_2, x_3, \ldots$ are retrieved. Then virtual token embeddings are inserted among discrete token embeddings and passed together into the PLM. During the fine-tuning phase, based on a task-specific loss, the parameters of the prompt encoder, $h_k$, are often tuned: $\mathcal{L}(x, y; h_k) = Loss(D(v \oplus x), y)$, where $D$ can be a decoder that maps the internal representation to task outputs, and *Loss* is an appropriate loss function,

such as Cross-Entropy Loss. During the fine-tuning, the PLM's parameters keep frozen, whereas the prompt encoder's parameters $h_k$ are updated in accordance with the loss.

## 3.2 FEDERATED PROMPT AVERAGING

In every communication round $t$, Federated Prompt Averaging includes the following steps.

**Initialization**: The server initializes the global prompt encoder $h_g^t$. Each client initializes its local prompt encoder $h_0^t, h_1^t, \ldots h_k^t$.

**Client Selection**: We select a fraction $C$ of the total $K$ clients for training. This subset size is $m = \max(C \times K, 1)$. The subset we choose is denoted as $S$.

**Local Encoder Tuning**: Each client $k$ in $S$ fetches the current global prompt encoder $h_g^t$, and assembles it with the PLM. During the local training on the local data $\mathcal{D}_k$, The PLM's parameters stay fixed while local prompt encoder parameters $h_k^t$ are tuned.

**Aggregation**: The server aggregates updates from all clients using weighted average. The global prompt encoder $h_g^{t+1}$ is updated based on the received parameters $h_k^t$ from clients for the next round of federated prompt tuning: $h_g^{t+1} = \sum_{k=1}^{K} \frac{|\mathcal{D}_k|}{\sum_{k=1}^{K}|\mathcal{D}_k|} h_k^t$.

## 4 EXPERIMENTAL SETUP

### 4.1 TASKS AND DATASETS

We evaluate our model using the popular XGLUE benchmark (Liang et al., 2020), a cross-lingual evaluation benchmark for our multilingual evaluation. We conduct our experiments on classification tasks of News Classification (NC), XNLI (Conneau et al., 2018) and MasakhaNEWS (Adelani et al., 2023). Accuracy (ACC) of the multi-class classification is used as the metric for both of the tasks. The details regarding each dataset can be found in Appendix D. Our base model for both tasks is the XLM-RoBERTa base-sized model (270M parameters), shown to perform well across many languages (Conneau et al., 2020).

### 4.2 MULTILINGUAL FINE-TUNING PARADIGMS

1) **Local Monolingual fine-tuning:** Traditional fine-tuning where a separate model is fine-tuned using the corresponding dataset for each single language locally. 2) **Centralized fine-tuning:** Standard fine-tuning using a combined dataset of all languages centralized in the cloud. 3) **Federated Full fine-tuning:** Directly fine-tuning the whole pre-trained language model in a federated manner, with the full pre-trained model on the server or each client. 4) **Federated Prompt Tuning:** Only training the prompt encoder in a federated manner, with the prompt encoder on the Server or each client. 5) **Federated LoRA (Low-Rank Adaptation):** Only training over-parameterized models with a low intrinsic rank in a federated manner, with the trainable rank decomposition matrices on the server or each client. In our tables, we use *PE* to denote the methods with parameter-efficient techniques.

## 5 EVALUATION AND ANALYSIS

### 5.1 MAIN RESULTS

Table 3 presents the experimental results on news classification. When employing parameter-efficient fine-tuning in comparison to full parameter fine-tuning, there is an acceptable decline in accuracy. Despite this decrease, the overall performance remains consistent and stable. A significant gain in accuracy is observed when adopting the FL approach. It is worth noting that the fine-tuning time is considerably reduced when employing the parameter-efficient method as opposed to without it. For a comprehensive analysis of this, refer to the section 5.4.

Table 2 summarises the results of our FL experiments on the XNLI task. To demonstrate the advantage of our Federated Prompt Tuning approach, a comparison was made with traditional monolingual training. As the result shows, our Federated Prompt Tuning, particularly on Non-IID

Table 1: Results for FL experiments on the NC task. Bold scores indicate the best in the column.

| Method | en | es | fr | de | ru | Avg |
|---|---|---|---|---|---|---|
| Monolingual | 92.4 | 84.7 | 79.5 | 88.3 | 89.0 | 86.8 |
| Centralized | 93.9 | 86.7 | **82.9** | **89.5** | 88.6 | 88.3 |
| FL (IID) | **94.1** | **86.9** | 82.7 | 89.4 | **88.8** | **88.4** |
| FL (Non-IID) | 92.4 | 86.3 | 81.2 | 88.9 | 84.7 | 86.7 |
| PE_Monolingual | 82.9 | 59.7 | 47.3 | 71.4 | 60.0 | 64.3 |
| PE_Centralized | 89.1 | 76.2 | 67.4 | 78.8 | 75.9 | 77.5 |
| PE_FL (IID) **(Ours)** | 91.2 | 82.2 | 76.5 | 86.4 | 81.6 | 83.6 |
| PE_FL (Prompt Tuning) (Non-IID) **(Ours)** | 87.8 | 79.2 | 73.7 | 83.1 | 79.5 | 80.7 |
| PE_FL (LoRA) (Non-IID) **(Ours)** | 89.3 | 76.0 | 75.4 | 75.8 | 83.2 | 79.9 |

Table 2: Results for FL experiments on the XNLI task. Bold scores indicate the best in the column. The PE_FL is evaluated under the Non-IID setting.

| Method | en | fr | es | de | el | bg | ru | tr | ar | vi | th | zh | hi | sw | ur | Avg |
|---|---|---|---|---|---|---|---|---|---|---|---|---|---|---|---|---|
| PE_Monolingual | 39.1 | 35.1 | 36.6 | 35.7 | 35.3 | 35.9 | 35.5 | 26.2 | 32.1 | 31.7 | 31.5 | 33.7 | 31.6 | 26.0 | 28.1 | 32.94 |
| PE_Centralized | 35.3 | 36.9 | 33.3 | 35.3 | 30.5 | 36.5 | 33.7 | 35.7 | 33.3 | **40.1** | 36.1 | 30.5 | 37.3 | **38.6** | 29.3 | 34.86 |
| PE_FL **(Ours)** | **43.2** | **40.6** | **42.9** | **40.2** | **39.7** | **40.8** | **41.1** | **37.6** | **39.1** | 39.9 | **39.4** | **39.8** | **38.2** | 37.1 | **37.8** | **39.83** |

Table 3: Results for FL experiments on MasakhaNEWS. Bold scores indicate the best in the column.

| Method | eng | fra | hau | swa | yor | Avg |
|---|---|---|---|---|---|---|
| PE_Monolingual | 79.08 | 84.91 | 75.18 | 76.64 | 52.8 | 73.7 |
| PE_Centralized | 79.81 | 87.10 | **80.78** | 84.18 | 64.48 | 79.3 |
| PE_FL (Prompt Tuning) **(Ours)** | 82.99 | **89.81** | 65.96 | **86.16** | 57.20 | 76.4 |
| PE_FL (LoRA) **(Ours)** | **87.10** | 85.64 | 76.64 | 83.21 | **72.50** | **81.0** |

setting, consistently outperformed the monolingual method across all languages. Remarkably, this superior performance was maintained even for languages with limited available data. The average accuracy further shows the advantage of Federated Prompt Tuning, marking a noticeable improvement from 32.94% in the monolingual approach to 39.83% with Non-IID Federated Prompt Tuning.

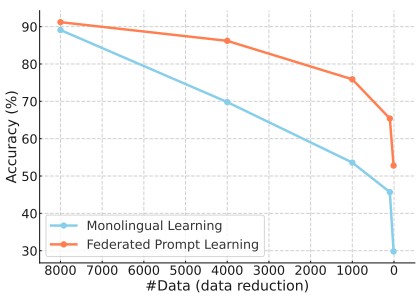

Figure 4: Performance comparison between traditional local fine-tuning and our federated prompt tuning method.

From our results in section 5.1, we observe that some languages demonstrate superior accuracy with the FL method compared to the centralized approach. This enhanced performance might be attributed to the Federated Prompt Averaging in FL, which could introduce similar implicit regularization effects (Izmailov et al., 2018; Rehman et al., 2022). Additionally, the prompt encoder, by freezing the core language model parameters, prevents the model from altering its foundational understanding of language. As a result, the model's tendency to overfitting is reduced, minimizing the risk of memorizing specific lexical cues and spurious correlations.

## 5.2 Ablation Study I: Data Efficiency

As previous sections mentioned, one characteristic of low-resource languages is their limited available data. Hence, enhancing data sample efficiency is crucial when fine-tuning pre-trained models for downstream tasks. To better validate and simulate the advantages of our approach in real-world scenarios, we reduced the data volume for one language and observed the performance under traditional local fine-tuning as well as our Federated prompt fine-tuning method. We conducted experiments on German News Classification. German was chosen because it represents the language with the fewest resources among the five languages included in this task.

As shown in the Figure 4, our Federated Prompt Tuning method consistently outperforms the traditional monolingual approach. As we reduce the dataset size from 8,000 to 30, the accuracy of

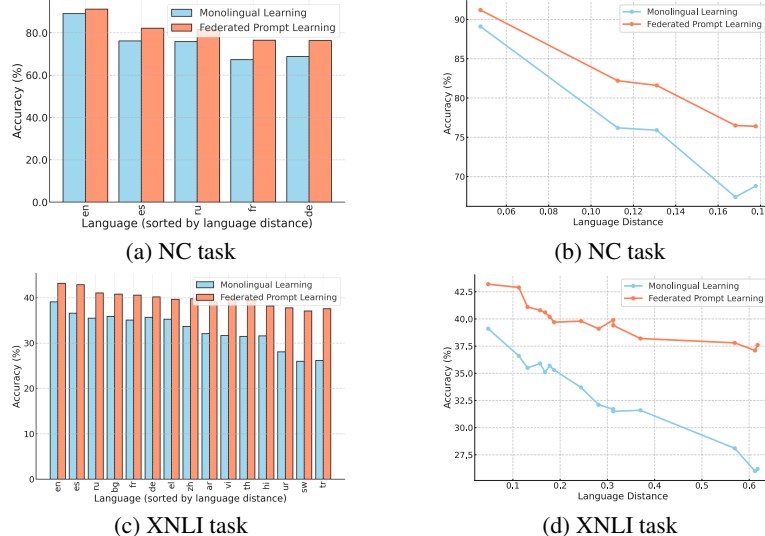

Figure 5: Comparative performance for both XNLI and NC tasks. (a)(c) reports the fine-tuning accuracy across different languages; (b)(d) reports the fine-tuning accuracy across languages with varying similarity to the pre-trained languages.

the traditional method drops significantly. On the other hand, the Federated Prompt Tuning method retains its performance, demonstrating its robustness even with limited data. This clearly indicates that our Federated Prompt Tuning approach is better suited for scenarios with limited data availability.

## 5.3 ABLATION STUDY II: LANGUAGE DISTANCE

As previously mentioned, another characteristic of low-resource languages is that their linguistic features differ from those of high-resource languages, particularly in aspects including syntax, phonology, and inventory. Consequently, direct fine-tuning on models pre-trained with highly dissimilar languages often yields unsatisfying results. Therefore, we conducted an ablation study to examine the impact of language similarity on performance, comparing our Federated Prompt fine-tuning method to the traditional local fine-tuning approach.

### 5.3.1 MULTILINGUAL DISTANCE MEASUREMENT

We define the *pre-trained language* as a representative composite language formed by blending each language in the multilingual corpus used for pre-training, in proportion to their amount. This is a formal representation for the mixed dataset composition. We define distance for a specific language in the downstream tasks, in terms of the negative logarithm of its similarity to the pre-trained language.

We leverage the database from Littell et al. (2017); Malaviya et al. (2017) to extract feature vectors for each language. These vectors are then weighted according to the token count of each language in the pre-trained corpus to calculate the feature vector of the pre-trained language. Given the feature vector $V_i$ for the $i$-th language, token count $T_i$, and total tokens $T_{\text{total}}$, the weight $w_i$ is given by $w_i = \frac{T_i}{T_{\text{total}}}$ and the feature vector $V_p$ for the pre-trained model is computed as $V_p = \sum_{i=1}^{n} w_i \cdot V_i$.

We define distance for a specific language in the downstream tasks, in terms of the negative logarithm of its cosine similarity to the pre-trained language. Let $v$ represent the feature vector of a specific language in the downstream task. The diversity measure $\phi$ between this language and the average language of the pre-trained model is defined as $\phi(v_i) = -\log(\cos(v_i, V_{\text{p}}))$.

### 5.3.2 FINE-TUNING LANGUAGES DISTANCE FROM PRE-TRAINED LANGUAGE

Leveraging the distance metric, we compared model performance of languages with varying degrees of distance to the pre-trained language. We present our results from two key experiments on the NC and XNLI tasks. From Figure 5, a conspicuous trend is observed: As the language similarity to

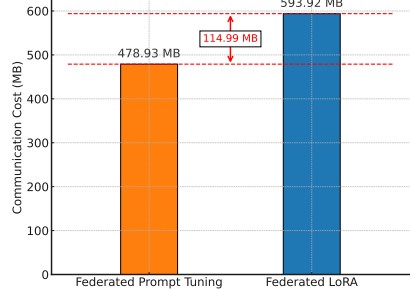

| | # Trainable Params | Communication Cost |
|---|---|---|
| Full fine-tuning | 278,655,764 | 110,592 MB |
| Prompt Tuning (Ours) | 1,202,708 | 479 MB |
| LoRA (Ours) | 1,491,476 | 594 MB |

Table 4: Comparison of parameter efficiency and communication overhead in NC task.

Figure 6: Communication Cost Comparison between Federated Prompt Tuning and Federated LoRA.

the pre-trained language decreases, the model's accuracy tends to drop. However, when we apply our Federated Prompt method, this decline is notably less steep. This means that even when we are dealing with languages that are quite different from the pre-trained one, our method manages to retain a decent level of accuracy. The difference between our method and the traditional local fine-tuning becomes even more obvious for languages with less data, indicating that our Federated Prompt Tuning method offers significant advantages, particularly in low-resource scenarios.

### 5.4 ABLATION STUDY III: PARAMETER EFFICIENCY

We evaluate the efficiency both in terms of computation and communication. From the perspective of trainable parameters, our method demonstrates exceptional parameter efficiency. In both tasks, despite the total number of parameters exceeding 278 million, the trainable parameters are only around 1.2 million, accounting forless than 0.5% of the total. Such design can substantially reduce training time and computational resources, while mitigating the risk of overfitting. Also, high parameter efficiency offers the potential for model deployment in resource-constrained environments.

Regarding communication, XLM-Roberta-Base's data transmission in FL with 5 clients and 10 communication rounds was 108 GB. After our optimization, using a prompt encoder with a $2 \times 768$ structure, the transmission size reduced to 478.93 MB, a 99% reduction shown in Table 4. This optimization enhances efficiency in federated prompt tuning and expands its applicability to bandwidth-constrained environments including edge devices and mobile networks.

## 6 CONCLUSION

Addressing the complexities of multilingual LLMs, especially for low-resource languages, requires innovative approaches that can balance efficiency, privacy concerns, and performance. Our Multilingual Federated Prompt Tuning paradigm provides a solution to these challenges. By aggregating lightweight multilingual prompts, this approach offers enhanced fine-tuning capabilities with minimal computational demand. The robustness of our method is especially pronounced for low-resource languages with sparse data and rare linguistic features. Overall, this approach promises to advance privacy and linguistic diversity in the realm of NLP. Future work will focus on exploring the impact on the Multilingual Federated Tuning method based on prompt learning as the model scale increases.

**Limitation** In our current paradigm, we have not added extra privacy protection techniques to defend against potential privacy attacks, such as gradient inversion (Geiping et al., 2020; Huang et al., 2021). We need to introduce differential privacy (Wei et al., 2020), secure aggregation (Bonawitz et al., 2016), and/or other methods to protect the training data. While FL by itself offers some privacy benefits, we understand it still needs more methods and attention to be really privacy-preserving, and we're interested in these limitations being addressed in follow-up work.

ETHICS STATEMENTS

This paper presents work whose goal is to advance the field of decentralized language model training and multilingual NLP. There are many potential societal consequences of our work, none of which we feel must be specifically highlighted here.

ACKNOWLEDGEMENT

This work was supported by the European Research Council via the REDIAL project (805194). We would also like to thank David Ifeoluwa Adelani, Shaozuo Yu and the anonymous reviewers for the insightful discussions and useful suggestions for the camera-ready version.

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

APPENDIX

## A  CHALLENGES AND OPPORTUNITIES OF MULTILINGUAL NLP

As natural language processing technologies advance, not all languages have been treated equally by developers and researchers. There are around 7,000 languages spoken in the world, and approximately 400 languages have more than 1 million speakers. However, there is scarce coverage of multilingual datasets. This is especially true for low-resource languages, where data scarcity is a major bottleneck. Furthermore, the under-indexing of certain languages is also driven by access to compute resources. Mobile data, compute, and other computational resources may often be expensive or unavailable in regions that are home to under-represented languages. Unless we address this disproportionate representation head-on, we risk perpetuating this divide and further widening the gap in language access to new technologies. One pressing example is biomedical data. Due to its global scale, this digital content is accessible in a variety of languages, yet most existing NLP tools remain English-centric. This situation highlights the need for effective strategies: how can we exploit abundant labeled data from resource-rich languages to make predictions in resource-lean languages?

Also, the problem is very timely compared to other application scenarios. It was not even considered a year ago. Previously, due to the smaller size of language models, the demand for data was not as high, and different kinds and sources of data were treated equally. Currently, the progress of LLMs, their usability, the amount of attention they receive, and the increased regulation on data, compound and lead to the urgency of this problem, where we are among the first batch to attempt to break both lingual and physical barriers.

## B  PROMPT CONSTRUCTION AND INITIALIZATION

When we use Prompt Tuning to optimize the parameter efficiency, the prompt tuning initialization text is *Predict the category given the following news article* for all the News Classification tasks. By providing the string of words, we initialize virtual token embeddings from existing embedding weights. This string is tokenized and tiled or truncated to match the number of virtual tokens.

## C  HYPERPARAMETERS AND IMPLEMENTATION

For all of the experiments, we report results using the 1e-3 learning rate and early stopping (5 epochs of no improvement). For a fair comparison with the setup in Houlsby et al. (2019b), we restrict the model sequence length to $512$ and use a fixed batch size for all tasks. For FL experiments, we adjust the parameter $\alpha$ that controls the mixture of languages in the dataset. An $\alpha$ value of 1.0 signifies a uniform mixture of all languages, while values closer to $0$ indicate a dominant representation of individual languages or a more separated mixture. When we use Prompt Tuning to optimize the parameter efficiency, the number of virtual tokens is $1$, and the prompt tuning init text is *Predict the category given the following news article* for all the News Classification tasks. When we use LoRA to optimize the parameter efficiency, We use two different ranks ($1$ and $8$), LoRA $\alpha$ is $16$ and LoRA dropout is $0.1$.

We use Hugging Face's transformers library (Wolf et al., 2020) and PEFT library (Mangrulkar et al., 2022) for loading pre-trained models and prompt tuning configurations. For our federated training and evaluation, we use the Flower framework (Beutel et al., 2020; Zhao et al., 2022) and PyTorch as the underlying auto-differentiation framework (Paszke et al., 2019). We use the AdamW optimizer (Loshchilov & Hutter, 2019; Kingma & Ba, 2015) for all experiments. All experiments are conducted using NVIDIA A40.

## D  DATASETS FOR GENERATIVE TASKS

**UN Corpus (Ziemski et al., 2016)** is a Machine Translation dataset of official records from the UN proceedings over the years 1990 to 2014, covering six languages: English, French, Spanish, Russian, Chinese, and Arabic. we sample 10k in each direction for training and 5k each for evaluation sets.

We cover three machine translation directions: En → Fr, Ar → Es, Ru → Zh, and sample 10k in each direction for training and 5k each for evaluation sets.

**News Classification (NC)** is a classification problem with 10 classes across 5 languages: English, Spanish, French, German, and Russian. This task aims to predict the category given a news article. Since only 10k annotated examples are available for each language (excluding the official test set), we sample 8k instances for training and 1k for evaluation sets.

**Cross-lingual Natural Language Inference (XNLI)** is a cross-lingual sentence understanding problem which covers 15 languages, including high-resource languages (English, French, Spanish, German, Russian and Chinese), medium-resource languages (Arabic, Turkish, Vietnamese and Bulgarian), and low-resource languages (Greek, Thai, Hindi, Swahili and Urdu). The task involves determining the relationship between a premise and a hypothesis sentence, and this relationship can be categorized into one of three classes: entailment, contradiction, or neutral. We sample 2k instances for training and 250 for evaluation sets for each language. NLI serves as an effective benchmark for assessing cross-lingual sentence representations, and better approaches for XNLI will lead to better general Cross-Lingual Understanding (XLU) techniques.

**MasakhaNEWS** is a benchmark dataset for news topic classification covering 16 languages widely spoken in Africa, where African languages are severely under-represented in NLP research due to lack of datasets covering several NLP tasks. The task involves categorizing news articles into different categories like sports, business, entertainment, and politics.We choose English, Hausa, Kiswahili, French and Yorùbá in our experiments. We sample 1433 instances for training and 411 for evaluation sets for each language.

## E  FEDERATED PROMPT AVERAGING ALGORITHM

---

**Algorithm 1** Federated Prompt Averaging

---

1: **Server executes:**
2: Initialize $h_g$
3: **for each** round $t$ **do**
4:     Select subset $S$ of $m$ clients
5:     **for each** client $k$ in $S$ **do**
6:         Send $h_g$ to client $k$
7:     **end for**
8:     Aggregate client updates:
9:     $h_g^{t+1} = \sum_{k=1}^{K} \frac{|\mathcal{D}_k|}{\sum_{k=1}^{K}|\mathcal{D}_k|} h_k^t$
10: **end for**

---

1: **Client $k$ executes:**
2: Retrieve current $h_g$
3: Assemble full model using $h_k$ and PLM parameters
4: Train model on $\mathcal{D}_k$
5: Update local prompt encoder $h_k$
6: Send updated $h_k$ to server

---

