# OpenReview forum: "Breaking Physical and Linguistic Borders: Multilingual Federated Prompt Tuning for Low-Resource Languages"
_ICLR.cc/2024/Conference — ICLR 2024 poster_

### Official Review · Reviewer_xDut · 2023-10-27

**Soundness:** 4 excellent
**Presentation:** 4 excellent
**Contribution:** 3 good
**Rating:** 8
**Confidence:** 5

**Summary:**

This paper is about multilingual federated prompt tuning for low-resource languages, bringing together federated learning and prompt-tuning techniques. This approach leverages parameter-efficient fine-tuning which preserves user privacy, and additionally, the authors introduce language distance in order to highlight the strengths of the proposed paradigm. The results show that the technique is parameter efficient and computationally beneficial, reducing by 99% the number of trainable parameters while increasing the performance on downstream tasks (XNLI, NC) of ~7% accuracy.

**Strengths:**

This paper makes a contribution to the federated learning field showing how federated learning can be used to enhance the performance of language models while preserving user privacy. The experiments are well-designed and the results are convincing - added to extensive analyses in order to leverage the capabilities of the proposed paradigm, but also its limitations.

**Weaknesses:**

Although the paper is generally well-structured, the title mentions `low-resource` languages. However, the two tasks leveraged are primarily on high-resource languages, rather than low-resourced language. I would suggest to the authors to include more tasks - there are many low-resource language datasets (for instance on African languages MasakhaNEWS, Masakhaner (1.0 and 2.0 - which have been cited by the way but not used), MasakhaPOS; Indic languages: https://github.com/AI4Bharat/indicnlp_catalog; etc) and tasks.

This is rather a highly recommended suggestion, that does not take away the contribution of the paper. Including them would strengthen the paper and be more in accordance with the title.

**Questions:**

The Aggregation formula is a bit confusing. Did you mean h_{global, t+1} = \sum_{k=1}^{m} h_{k, t}? Because the `t+1` on the last term does not make sense to me.

---

> ### Author Response · Authors · 2023-11-20
> **Response to Reviewer xDut**
>
> ###
>
> We thank the reviewer for the insightful and positive feedback!
>
> **W1**
>
> > the two tasks leveraged are primarily on high-resource languages, rather than low-resourced language. I would suggest to the authors to include more tasks - there are many low-resource language datasets (for instance on African languages MasakhaNEWS, Masakhaner (1.0 and 2.0 - which have been cited by the way but not used), MasakhaPOS; Indic languages: https://github.com/AI4Bharat/indicnlp_catalog; etc) and tasks.
>
> Thank you for recommending these excellent datasets for our evaluation.
>
> We agree that diversifying our dataset to include African and Indic languages will significantly strengthen our paper's scope and alignment with its title. To address this, we have initiated experiments with MasakhaNEWS and plan to conduct further research with MasakhaNER and IndicNLP Catalog datasets shortly.
>
> **MasakhaNEWS** is a benchmark dataset for news topic classification covering 16 languages widely spoken in Africa, where African languages are severely under-represented in NLP research due to lack of datasets covering several NLP tasks. The task involves categorizing news articles into different categories like sports, business, entertainment, and politics. We choose English, Hausa, Kiswahili, French and Yorùbá in our preliminary experiments. We sample 1433 instances for training and 411 for evaluation sets for each language.
>
> Table 2 in our revised paper presents the preliminary results of experiments focused on MasakhaNEWS. When employing Federated Prompt Tuning in comparison to the monolingual and centralized counterparts, in general, a significant gain in accuracy is observed when adopting our Federated approach compared with the monolingual baseline.
>
> | Method                        | eng   | fra   | hau   | swa   | yor   | Avg  |
> |-------------------------------|-------|-------|-------|-------|-------|------|
> | PE_Monolingual                | 79.08 | 84.91 | 75.18 | 76.64 | 52.8  | 73.7 |
> | PE_Centralized                | 79.81 | 87.10 | 80.78 | 84.18 | 64.48 | 79.3 |
> | PE_FL (Ours)  | 82.99 | 89.81 | 65.96 | 86.16 | 57.20 | 76.4 |
>
> Q1
>
> > The Aggregation formula is a bit confusing. Did you mean h_{global, t+1} = \sum_{k=1}^{m} h_{k, t}? Because the `t+1` on the last term does not make sense to me.
>
> We appreciate your detailed feedback on the notation! We have revised the formula on page 5 and added more details to avoid any confusion in our updated version.

---

> > ### Comment · Reviewer_xDut · 2023-11-20
> > **Feedback**
> >
> > Thanks for the new results and integration. Can you please also report the experiments on African languages ?

---

> > > ### Comment · Reviewer_AG4r · 2023-11-21
> > > **English and French are not low resource languages**
> > >
> > > Also, how to make sense of the fact that in PE_Centralized, low resource languages like hausa and swahili outperformed english?

---

> > > > ### Author Response · Authors · 2023-11-23
> > > > **Response to Follow-up Questions**
> > > >
> > > > Thank you for your follow-up questions!
> > > >
> > > > The possible reason why low resource languages like Hausa and Swahili outperformed English in centralized finetuning is because of the unique experimental setting for our preliminary experiments. Instead of using the entire original MasakhaNEWS dataset, we selected five languages with the most data samples and kept the data samples the same for each language, which means that, during finetuning, the combined data volume of these low-resource languages exceeded that of the English corpus. We are currently conducting follow-up experiments using the full dataset across all languages and conducting repeated trials. We will report our findings once these experiments are complete.

---

### Official Review · Reviewer_E7Lk · 2023-10-28

**Soundness:** 1 poor
**Presentation:** 2 fair
**Contribution:** 2 fair
**Rating:** 1
**Confidence:** 5

**Summary:**

The paper introduces a finetuning paradigm that combines federated learning (FL) with prompt tuning for multilingual finetuning on certain, with the goal to preserve the privacy of the local data used for the finetuning job. The results show better performance in certain classification tasks, such as New Classification and XNLI.

**Strengths:**

- Federated learning have recently gained good traction, the paper is a good application of it in the tasks of finetuning LLM. The paper chooses to use prompt tuning instead of full tuning to save costs, as well as to avoid overfitting on small data.
- The method produces better performance on the 2 classification tasks compared to baselines

**Weaknesses:**

- The proposed is a very trivial combination of federated learning and prompt tuning, which both are established methodology in their own realm. There is no novelty, such as modification or adjustment to the method that may have give a better results. In other words, people with an objective to do federated learning for privacy purpose can easily come up with prompt tuning as a solution to reduce costs.
- Though it may have implicitly inferred by the concept of FL, the paper did not mention why and how federated learning helps with privacy and in which case one should use FL for their application.
- The purpose of the task of multilingual finetuning in this case, is not warranted use case of privacy preservation.
- There is no reported evidence that privacy is actually preserved. Such as whether the final model memorize the local data.
- There are better parameter-efficient finetuning methods, such as LORA/QLora, that the authors should conduct experiments on and do comparision with prompt tuning.
- The results show prompt tuning are much worse than full-federated tuning, thus casting doubt if the cost-saving is worth it.
- Other generative and knowledge-based tasks, such as QA, translations and summarizations should be performed.

**I have read the author responses and I advocate for a strong reject, below are reasons:**

* I mentioned the paper has fundamental problems with originality, novelty, where the paper uses an unrelated existing and non-novel method designed for a different problem (fed-learning) to solve a low-resource "privacy" problem that does not make sense or exist yet, in which the method itself much worse than standard training.
* Instead of addressing the scientific issue, the authors distracted away by pressing that they are helping the low-resource communities, or improving inequality as a societal issue. These multiple responses are lengthy, wordy, unnecessary, and filled with many "politically correct" (I don't know better word) things to avoid the scientific issue. Agree that we should help those under-represented communities, but after reading these, I shouldn't feel like rejecting the paper is an action against those communities.
* The problem of "a low-resource community who wants to shut down their internet and border" is unfounded. We train LLM on public data we can find. If they wants to protect their secret data, they can download a public pre-trained model and fine-tune on their own.
* The real problem is how to improve low-resource with the limited data we have, which the paper fails to suggest a better solution than trivial.
* Less communication doens't mean more privacy, because we transfer model weights, not the data. And less parameters doesn't mean less private information be leaked. This misconception leads to wrong approach.
* The author claims to be the first to target the low-resource problem and many other things, but there have been many works in previous years about this. Please be careful with this kind of "we are first" statements.
* Overall, none of the responses has helped resolve the issues stated in the review.

**Questions:**

- Citation formet incorrect, \citep{} be used to produce something like (Abc, et al., 2023) and not Abc, et al., 2023 everywhere.
- Many grammatical errors, such as "Throughout the fine-tuning...""

---

> ### Author Response · Authors · 2023-11-22
> **Response to Reviewer E7Lk [1/6]**
>
> ## Q1
> > Citation formet incorrect, \citep{} be used to produce something like (Abc, et al., 2023) and not Abc, et al., 2023 everywhere.
>
> Thanks for pointing it out. We have corrected the citations for all references in our revised paper.
>
> ## Q2
> > Many grammatical errors, such as "Throughout the fine-tuning...""
>
> We appreciate your feedback on the grammatical errors. We have revised the grammar to avoid any confusion in our updated version.
>
> ## W4
> > Other generative and knowledge-based tasks, such as QA, translations and summarizations should be performed.
>
> We appreciate the feedback. Our current paradigm is general-purpose and can be easily adapted to other generative and knowledge-based tasks. In response, we have expanded our evaluations to encompass a broader range of scenarios, addressing the concern of limited task selection. This rebuttal is part of a series, and we will provide additional results during the discussion period.
>
>
> * Low-resource Dataset
>
> Our setting and results: **MasakhaNEWS**, covering 16 languages widely spoken in Africa, where African languages are severely under-represented in NLP research due to a lack of datasets covering several NLP tasks. The task involves categorizing news articles into different categories like sports, business, entertainment, and politics. We chose English, Hausa, Kiswahili, French and Yorùbá in our preliminary experiments. We sample 1433 instances for training and 411 for evaluation sets for each language.
>
> Table 2 in our revised paper presents the preliminary results of experiments focused on MasakhaNEWS. When employing Federated Prompt Tuning in comparison to the monolingual and centralized counterparts, in general, a significant gain in accuracy is observed when adopting our Federated approach compared with the monolingual baseline.
>
> | Method | eng | fra | hau | swa | yor | Avg |
> | --- | --- | --- | --- | --- | --- | --- |
> | PE_Monolingual | 79.08 | 84.91 | 75.18 | 76.64 | 52.8 | 73.7 |
> | PE_Centralized | 79.81 | 87.10 | 80.78 | 84.18 | 64.48 | 79.3 |
> | PE_FL (Ours) | 82.99 | 89.81 | 65.96 | 86.16 | 57.20 | 76.4 |
>
> * Question Answering
>
> Dataset and our setting: **MultiLingual Question Answering (MLQA)** is a benchmark dataset for cross-lingual question-answering performance, covering English, Arabic, German, Spanish, Hindi, Vietnamese and Simplified Chinese. We sample 1433 instances for training and 411 for evaluation sets for each language.
>
> * Machine Translation
>
> Dataset and our setting: **UN Corpus** is a Machine Translation dataset of official records from the UN proceedings over the years 1990 to 2014, covering six languages: English, French, Spanish, Russian, Chinese, and Arabic. we sample 10k in each direction for training and 5k each for evaluation sets. We cover three machine translation directions: En → Fr, Ar → Es, Ru → Zh, and sample 10k in each direction for training and 5k each for evaluation sets.
>
> Once we get the result, we’ll update our response and our paper.

---

> ### Author Response · Authors · 2023-11-22
> **Response to Reviewer E7Lk [2/6]**
>
> ## W1
> > The proposed is a very trivial combination of federated learning and prompt tuning, which both are established methodology in their own realm. There is **no novelty, such as modification or adjustment to the method that may have give a better results**. In other words, people with an objective to do federated learning for privacy purpose can easily come up with prompt tuning as a solution to reduce costs.
>
> We appreciate the opportunity to address the concerns raised by the reviewer and would like to defend our proposal, emphasizing its novelty and significance. In summary, we would like to clarify that our paper introduces federated prompt tuning as a solution to help address the **linguistic and geographic boundaries** hindering the application of LLMs to **various regions and lower-resource languages**.
>
> We would like to further clarify from the following two aspects:
>
> 1. We emphasize the unique background and pressing need of our work, as we noticed the review may **overlook the part of multilingual and low-resource language in the initial review**.
>
> First, our work primarily focuses on the under-representation of multilingual and low-resource languages in large language models. As natural language processing technologies advance, not all languages have been treated equally by developers and researchers. There are around 7,000 languages spoken in the world, and approximately 400 languages have more than 1 million speakers. However, there is scarce coverage of multilingual datasets. This is especially true for low-resource languages, where data scarcity is a major bottleneck. Furthermore, the under-indexing of certain languages is also driven by access to compute resources. Mobile data, compute, and other computational resources may often be expensive or unavailable in regions that are home to **under-represented languages**. Unless we address this disproportionate representation head-on, we risk perpetuating this divide and further widening the gap in language access to new technologies. One pressing example is **biomedical data**. Due to its global scale, this digital content is accessible in a variety of languages, yet most existing NLP tools remain English-centric. This situation highlights the need for effective strategies: how can we exploit abundant labeled data from resource-rich languages to make predictions in resource-lean languages?
>
> Also, we wanted to highlight the urgency and timeliness of the problem. The problem is very timely compared to other application scenarios. It was not even considered a year ago. Previously, due to the smaller size of language models, the demand for data was not as high, and different kinds and sources of data were treated equally. Currently, the progress of LLMs, their usability, the amount of attention they receive, and the increased regulation on data, compound and lead to the urgency of this problem, where we are among **the first batch to attempt to break both lingual and physical barriers**.

---

> ### Author Response · Authors · 2023-11-22
> **Response to Reviewer E7Lk [3/6]**
>
> 2. We emphasize the simplicity and effectiveness of our proposed paradigm, in which case **modification or adjustment are not necessary but might make it complex**.
>
> In previous cases, data transmission was always one-directional. Existing approaches focus on solving this locally, for example, through cross-lingual transfer, as well as data augmentation and preference training to address these bottlenecks.
>
> In our paper, we approach it from a collaborative (two-directional) perspective. By training LLMs collaboratively across multiple participants without sharing raw data, the accuracy, robustness, and generalizability of LLMs can be enhanced by **leveraging collective knowledge and exposing models to a wider range of linguistic patterns**.
>
> There exists very little research from such a collaborative perspective for low-resource languages. **Our findings open up new avenues for exploration and have the potential to inspire future research in this area**.
>
> **Following the principle of Occam's razor**, we adopt the concept of "federated" as a simple and established concept to describe our solution to the problem, which not only contributes a timely and practical solution to a rapidly evolving field but also vividly depicts the key innovation of our paradigm: knowledge sharing and aggregation without data transmission.
>
> Additionally, from the federated learning perspective, as far as we know, we are the **first paper to investigate the data efficiency and transferability** brought by federated learning, and we believe this sheds some light on how federated learning can benefit LLM training in terms of generalizability and stability, beyond simply mitigating compliance risks.
>
> [1] Joshi, Pratik, et al. The state and fate of linguistic diversity and inclusion in the NLP world. ACL 2020.
>
> [2] Bérard et al., A Multilingual Neural Machine Translation Model for Biomedical Data NLP-COVID19 2020.
>
> [3] Lauscher et al., From Zero to Hero: On the Limitations of Zero-Shot Language Transfer with Multilingual Transformers. EMNLP 2020.
>
> [4] Xia et al., Generalized Data Augmentation for Low-Resource Translation. ACL 2019.

---

> ### Author Response · Authors · 2023-11-22
> **Response to Reviewer E7Lk [4/6]**
>
> ## W2
> > Though it may have implicitly inferred by the concept of FL, the paper did not mention why and how federated learning helps with privacy and in which case one should use FL for their application. The purpose of the task of multilingual finetuning in this case, is not warranted use case of privacy preservation. There is no reported evidence that privacy is actually preserved. Such as whether the final model memorize the local data.
>
> We thank the reviewer for the insightful comments and concerns regarding privacy! We appreciate the opportunity to clarify this aspect of our work. It's important to note that multilingual finetuning here is not an approach for preserving privacy but rather a problem we aim to solve.
>
> 1. Our approach inherently supports data privacy, specifically by complying with international data privacy regulations. This compliance minimizes the need for cross-border data transmission, ensuring legal compliance and facilitating collaboration among entities with limited local computing resources, as detailed in Section 3.2.
>
> 2. We directly address privacy concerns by reducing the volume of transmitted data, thereby limiting potential privacy breaches. As demonstrated in Section 5.4, transmitting fewer parameters significantly reduces the risk of privacy leakage, aligning our methodology with the privacy focus highlighted in the abstract.
>
> 3. We would like to provide evidence for alleviating memorization from the following aspects:
>
> (1) By freezing the core language model parameters, we prevent the model from altering its foundational understanding of language. Consequently, the prompt encoder reduces the risk of memorizing specific lexical cues and spurious correlations, as discussed in Section 5.1 [1].
>
> (2) Components of Federated Learning play an essential role in reducing unintended memorization [2]. Specifically, clustering data according to users—a key design element in FL—significantly reduces such memorization. Additionally, using the Federated Averaging method for training further decreases the risk.
>
> 4. Regarding privacy protection, we acknowledge that we did not add extra privacy protection techniques to defend against potential privacy attacks, such as gradient inversion. Therefore, we appreciate the reviewer's valid points and have revised our paper to clarify our contribution to privacy and removed related claims about the capability of privacy protection to avoid confusion. However, various methods like secure aggregation (SA) and differential privacy (DP) can be applied in conjunction with our pipeline to further enhance privacy protection.
>
> [1] Lester, Brian, Rami Al-Rfou, and Noah Constant. "The Power of Scale for Parameter-Efficient Prompt Tuning." EMNLP 2021.
>
> [2] Thakkar et al. "Understanding Unintended Memorization in Language Models Under Federated Learning." PrivateNLP 2021.

---

> ### Author Response · Authors · 2023-11-22
> **Response to Reviewer E7Lk [5/6]**
>
> ## W3
> > There are better parameter-efficient finetuning methods, such as LORA/QLora, that the authors should conduct experiments on and do comparision with prompt tuning.
> > The results show prompt tuning are much worse than full-federated tuning, thus casting doubt if the cost-saving is worth it.
>
> Thank you for your valuable suggestions! Following the reviewer's constructive feedback, we have implemented experiments with LoRA (r=8, lora_alpha=16, lora_dropout=0.1) and summarized the results in the table below.
>
> Table 4 in our revised paper presents the preliminary results of experiments on the NC task. Bold scores indicate the best performance between Prompt Tuning and LoRA in each column.
>
> | Method                          | en   | es   | fr   | de   | ru   | Avg  |
> |---------------------------------|------|------|------|------|------|------|
> | Monolingual                     | 92.4 | 84.7 | 79.5 | 88.3 | 89.0 | 86.8 |
> | Centralized                     | 93.9 | 86.7 | 82.9 | 89.5 | 88.6 | 88.3 |
> | FL (IID)                        | 94.1 | 86.9 | 82.7 | 89.4 | 88.8 | 88.4 |
> | FL (Non-IID)                    | 92.4 | 86.3 | 81.2 | 88.9 | 84.7 | 86.7 |
> | PE_Monolingual                  | 82.9 | 59.7 | 47.3 | 71.4 | 60.0 | 64.3 |
> | PE_Centralized                  | 89.1 | 76.2 | 67.4 | 78.8 | 75.9 | 77.5 |
> | PE_FL (IID) (Ours)              | 91.2 | 82.2 | 76.5 | 86.4 | 81.6 | 83.6 |
> | PE_FL (Prompt Tuning) (Non-IID) (Ours) | 87.8 | **79.2** | 73.7 | **83.1** | 79.5 | **80.7** |
> | PE_FL (LoRA) (Non-IID) (Ours)   | **89.3** | 76.0 | **75.4** | 75.8 | **83.2** | 79.9 |
>
> We also conducted a comparison of parameter efficiency and communication overhead in the NC task:
>
> | Method                   | # Trainable Params | Communication Cost |
> |--------------------------|--------------------|--------------------|
> | Federated Full Finetuning| 278,655,764        | 108GB              |
> | Federated Prompt Tuning (Ours) | 1,202,708     | 478.93MB           |
> | Federated LoRA (Ours)    | 1,491,476          | 593.92MB           |
>
> Additionally, we have included Figure 7 in Section 5.4 of our revised paper to clearly illustrate the comparison between Prompt Tuning and LoRA.
>
> The results demonstrate that Federated LoRA and Federated Prompt Tuning achieve comparable performance, with Federated Prompt Tuning showing a slight advantage. In terms of data transmission and communication cost, Prompt Tuning requires only about 80% of the resources compared to LoRA, and leads to less privacy leakage.
>
> Furthermore, as the pretrained model scales up, the performance of Prompt Tuning rapidly improves, approaching or even surpassing full finetuning. This indicates its significant potential. Prompt Tuning also enhances the overall model's generalization capabilities. The prompt encoder acts as a mechanism to extract linguistic-specific patterns on the client and general linguistic patterns on the server, showcasing advantages that adapters cannot match.
>
> This justifies our use of Federated Prompt Tuning in our research, considering its efficiency in terms of parameters and communication, as well as its capability for generalization and adaptation to low-resource languages, which are crucial and undoubtedly worth it.
>
> Regarding QLoRA, we did not consider quantization in our study since our focus is solely on updating the parameters of prompt encoders on clients and the server, keeping the pre-trained model frozen. QLoRA involves quantizing the pre-trained models, which falls outside the scope of our discussion and does not contribute to reducing communication costs, a key bottleneck in the federated setting.

---

> ### Author Response · Authors · 2023-11-22
> **Response to Reviewer E7Lk [6/6]**
>
> please see our response to W5 in our first comment

---

> ### Author Response · Authors · 2023-11-23
> **Machine Tanslation results**
>
> We are pleased to present our preliminary results for Machine Translation. The process took more time than the classification tasks.
>
> Our pre-trained model is [M2M-100 with 418M parameters](https://huggingface.co/facebook/m2m100_418M), a many-to-many MT model capable of translating between any pairing of 100 languages.
>
> The table below displays our preliminary results on fine-tuning a machine translation model using the UN Corpus dataset. The scores are measured with sacreBLEU; higher scores indicate better performance.
>
> We can use the pre-trained model as a baseline without fine-tuning. In all cases, fine-tuning demonstrates improvement compared to using the pre-trained model directly. Additionally, our federated prompt tuning paradigm outperforms its centralized counterpart (33.4 avg. BLEU for Ours vs. 32.5 for Centralized), demonstrating the superiority of our proposed paradigm.
>
> | Method          | En-Fr | Ar-Es | Ru-Zh | Avg  |
> |-----------------|-------|-------|-------|------|
> | Pretrained Model     | 31.4  | 27.4  | 27.9  | 28.9 |
> | PE_Centralized  | 33.9  | 31.9  | 30.7  | 32.5 |
> | PE_PT (Non-IID) (Ours)| 34.3  | 33.8  | 32.1  | 33.4 |

---

> > ### Author Response · Authors · 2023-11-23
> > **Looking forward to your responses or further suggestions/comments!**
> >
> > Dear Reviewer E7Lk,
> >
> > We would like to sincerely thank you again for your time in reviewing our work!
> >
> > We understand you might be quite busy. However, as the discussion deadline is approaching, would you mind checking our response and confirming whether you have any further concerns or questions? Any further comments and discussions are welcomed!
> >
> > Best Regards,
> >
> > The authors of Paper1909

---

### Official Review · Reviewer_AG4r · 2023-10-31

**Soundness:** 4 excellent
**Presentation:** 1 poor
**Contribution:** 3 good
**Rating:** 3
**Confidence:** 4

**Summary:**

The paper proposes a Multilingual Federated Prompt Tuning paradigm, where lightweight multilingual prompts are encoded and on regional devices in different languages and aggregated by averaging the prompt embeddings. The goal is fine-tuning multilingual large language models on resource-constraint devices in a privacy-preserving way. The paper evaluates this approach via the XNLI task, ablated into data efficiency, "language distance", and communication cost, against "monolingual" training (baseline).

**Strengths:**

The innovation lies in that the paper somehow mashes federated learning, multi-lingual (low resource) language models, and Parameter-Efficient Fine-Tuning in one paper. The fact that they managed to come up with a storyline for a system that bolsters the benefit of each approach is commendable.

**Weaknesses:**

- poor presentation: the citations are not separable enough from the main text, e.g., without any parenthesis, rendering the submission unreadable. Against the tradition and ease of reading, abbreviations are not defined in advance, e.g., NLI, PFL, PLM.
- claims unverifiable: no code release.
- conflating existing metrics with innovation: language distance is not a new concept.
- conceptual weakness: the contrived baseline was bound to give the proposed approach an edge due to lack of federated learning. Also, what the paper refers to as prompts are just classifier model input, which are different from decoders-style LLM prompts as commonly acknowledged. Finally, the approach has absolutely nothing to do with privacy which the abstract and the main body consistently bolsters.
- evaluation weakness: only two tasks (new classification and XNLI) was used in evaluation.

**Questions:**

In section 5.4.1

>  In both the NC and XNLI tasks, despite the total number of
parameters exceeding 278 million, the trainable parameters are only around 1.2 million, accounting
for less than 0.5% of the total.

Could the authors clarify which part of the model is being fine-tuned?

---

> ### Author Response · Authors · 2023-11-21
> **Response to Reviewer AG4r [1/3]**
>
> ### Q
>
> > In section 5.4.1
> > In both the NC and XNLI tasks, despite the total number of parameters exceeding 278 million, the trainable parameters are only around 1.2 million, accounting for less than 0.5% of the total.
> > Could the authors clarify which part of the model is being fine-tuned?
>
> Yes, we clarify that we only update the prompt encoders. This includes the parameters of the local prompt encoders $h_k$ on Client k, and the parameters of the global encoder $h_g$ on the server in the revised paper (referred to as $h_{global}$ in the original paper). During this process, we keep the pre-trained language models frozen at all times.
>
> Therefore, the trainable parameters are solely those within the prompt encoders, in contrast to the total number of parameters involved in full fine-tuning.
>
> To further clarify and avoid any confusion, we have revised some details in Section 3. Additionally, we have attached a figure (Figure 2) in the revised version of the paper, illustrating the architecture of our prompt encoder and the tuning process.
>
> ### W1
> > poor presentation: the citations are not separable enough from the main text, e.g., without any parenthesis, rendering the submission unreadable. Against the tradition and ease of reading, abbreviations are not defined in advance, e.g., NLI, PFL, PLM.
>
> We apologize for any confusion caused by the current citation format. We have corrected the citations for all references in our revised paper.
>
> We realize the oversight in not defining certain abbreviations, such as NLI (Natural Language Inference), PFL (Prompt Federated Learning), and PLM (Pre-trained Language Models), at their first occurrence in the text. We appreciate you highlighting this point. In the revised paper, we have ensured that all abbreviations are defined upon their first use.
>
> ### W2
> > claims unverifiable: no code release.
>
> We provide an anonymized version of the code repository, accessible through this link: https://anonymous.4open.science/r/Breaking_Physical_and_Linguistic_Borders-F1C5.
>
> ### W3
> > conflating existing metrics with innovation: language distance is not a new concept.
>
> Thank you for your insightful comments on our paper.
>
> We acknowledge and agree with your review that the concept of language distance is not novel, having been explored in various contexts previously. However, we emphasize that our work introduces this concept within a unique and specific scenario: multilingual federated tuning. Our novel application provides a fresh perspective on language distance as a metric that illustrates the **transferability** relationships in multilingual NLP. This enables us to analyze the performance of federated prompt tuning and local monolingual transfer learning for low-resource languages.
>
>
> To avoid overemphasizing the novelty of the language distance concept itself, we've amended the language in the abstract from "introduce language distance as a new concept" to "present a new notion of language distance" in our revised paper. This modification more accurately reflects our contribution.
>
> We refer to [1] and [2] for our language distance measurement. Furthermore, we are open to learning about any additional references or works related to language distance that the reviewers may be aware of. If there are specific references that we have overlooked or that could further strengthen our work, we welcome their inclusion to enhance the comprehensiveness and depth of our research.
>
> [1] Malaviya et al., Learning Language Representations for Typology Prediction. *EMNLP 2017*.
>
> [2] Littell et al., URIEL and lang2vec: Representing languages as typological, geographical, and phylogenetic vectors. *EACL 2017*.

---

> ### Author Response · Authors · 2023-11-21
> **Response to Reviewer AG4r [2/3]**
>
> ## W4
>
> > conceptual weakness: the contrived baseline was bound to give the proposed approach an edge due to lack of federated learning. Also, what the paper refers to as prompts are just classifier model input, which are different from decoders-style LLM prompts as commonly acknowledged. Finally, the approach has absolutely nothing to do with privacy which the abstract and the main body consistently bolsters.
> >
>
> ### Regarding concerns about baselines
>
> We appreciate the opportunity to clarify this aspect of our work.
>
> In previous cases, data transmission was always one-directional. Existing approaches focus on solving this locally, for example, through local transfer with monolingual data.
>
> In our paper, we approach it from a collaborative perspective, which we call federated prompt tuning in our paper. By training LLMs collaboratively across multiple participants without sharing raw data, the accuracy, robustness, and generalizability of LLMs can be enhanced by leveraging collective knowledge and exposing models to a wider range of linguistic patterns.
>
> As you mentioned, there exists very little research from such a collaborative perspective for low-resource languages. **Our findings open up new avenues for exploration and have the potential to inspire future research in this area.**
>
> What we would like to demonstrate is not simply the performance boost. It’s the data efficiency (Section 5.2) and transferability for different language similarities (Section 5.3) of our paradigm’s superiority on low-resource languages.
>
> -----
>
> ### Regarding concerns about prompts
>
> We would like to clarify that the prompts in our paper are NOT the same as classifier model input, and they are suited for all decoders-style LLMs. To further clarify the prompt tuning procedure and the prompt construction, we've added more details in Section 3 and Appendix B, and C in the revised version.
>
> Instead of selecting discrete text prompts in a manual or automated fashion, in our paradigm, we utilize **virtual prompt** embeddings that can be optimized via gradient descent. Specifically, each prompt encoder, whether global or local, takes a series of virtual tokens, which are updated during tuning to better aid the model.
>
> Figure 2 in our revised paper shows how our prompt tuning works on both clients and the server. Specifically, a textual prompt tailored for a specific task and input text is passed to the model. The task-specific virtual tokens are retrieved based on the textual prompt. With the input text tokenized, the discrete word token embeddings are retrieved. Then virtual token embeddings are inserted among discrete token embeddings and passed together into the pretrained models. Therefore, **the prompt is adaptable across various pre-trained model architectures, including decoder-style, encoder-style, or encoder-decoder style.**
>
> ----
>
> ### Regarding concerns about privacy
>
> We appreciate the opportunity to clarify this aspect of our work.
>
> - Our approach inherently supports data privacy. Specifically, it complies with international data privacy regulations by minimizing the need for cross-border data transmission. This not only ensures legal compliance but also facilitates collaboration among entities with limited local computing resources, as detailed in section 3.2.
>
> - We directly address privacy concerns by reducing the volume of transmitted data, thereby limiting potential privacy breaches. As demonstrated in section 5.4, transmitting fewer parameters significantly reduces the risk of privacy leakage, aligning our methodology with the privacy focus highlighted in the abstract.

---

> > ### Author Response · Authors · 2023-11-21
> > **Response to Reviewer AG4r [3/3]**
> >
> > ### W5
> >
> > > evaluation weakness: only two tasks (new classification and XNLI) was used in evaluation.
> > >
> >
> > we would like to highlight additional evaluation results that we have been conducting to substantiate our claims further. These additional evaluations encompass a broader range of tasks and scenarios, which we believe address the concern of limited task selection. This response is the first in a series of comprehensive rebuttals; we will provide additional experimental results during the discussion period.
> >
> > **MasakhaNEWS** is a benchmark dataset for news topic classification covering 16 languages widely spoken in Africa, where African languages are severely under-represented in NLP research due to a lack of datasets covering several NLP tasks. The task involves categorizing news articles into different categories like sports, business, entertainment, and politics. We chose English, Hausa, Kiswahili, French and Yorùbá in our preliminary experiments. We sample 1433 instances for training and 411 for evaluation sets for each language.
> >
> > Table 2 in our revised paper presents the preliminary results of experiments focused on MasakhaNEWS. When employing Federated Prompt Tuning in comparison to the monolingual and centralized counterparts, in general, a significant gain in accuracy is observed when adopting our Federated approach compared with the monolingual baseline.
> >
> > | Method | eng | fra | hau | swa | yor | Avg |
> > | --- | --- | --- | --- | --- | --- | --- |
> > | PE_Monolingual | 79.08 | 84.91 | 75.18 | 76.64 | 52.8 | 73.7 |
> > | PE_Centralized | 79.81 | 87.10 | 80.78 | 84.18 | 64.48 | 79.3 |
> > | PE_FL (Ours) | 82.99 | 89.81 | 65.96 | 86.16 | 57.20 | 76.4 |
> >
> > The tasks we’re currently working on are as follows:
> >
> > - Question Answering
> >
> > Dataset and our setting: **MultiLingual Question Answering (MLQA)**  is a benchmark dataset for cross-lingual question-answering performance, covering English, Arabic, German, Spanish, Hindi, Vietnamese and Simplified Chinese. % We sample 1433 instances for training and 411 for evaluation sets for each language.
> >
> > - Machine Translation
> >
> > Dataset and our setting: **UN Corpus** is a Machine Translation dataset of official records from the UN proceedings over the years 1990 to 2014, covering six languages: English, French, Spanish, Russian, Chinese, and Arabic. we sample 10k in each direction for training and 5k each for evaluation sets. We cover three machine translation directions: En → Fr, Ar → Es, Ru → Zh, and sample 10k in each direction for training and 5k each for evaluation sets.
> >
> > Once we get the result, we’ll update our response and our paper.

---

> > > ### Comment · Reviewer_AG4r · 2023-11-22
> > > **Look forward to learning about the results**
> > >
> > > Look forward to learning about the results on Question Answering and Machine Translation.
> > > Updated soundness

---

> ### Author Response · Authors · 2023-11-23
> **Machine Tanslation results**
>
> We are pleased to present our preliminary results for Machine Translation. The finetuning took more time than the classification tasks.
>
> Our pre-trained model is [M2M-100 with 418M parameters](https://huggingface.co/facebook/m2m100_418M), a many-to-many MT model capable of translating between any pairing of 100 languages.
>
> The table below shows our preliminary results on fine-tuning a machine translation model using the UN Corpus dataset. The scores are measured with sacreBLEU; higher scores indicate better performance.
>
> We can use the pre-trained model as a baseline without fine-tuning. In all cases, fine-tuning demonstrates improvement compared to using the pre-trained model directly. Additionally, our federated prompt tuning paradigm outperforms its centralized counterpart (33.4 avg. BLEU for Ours vs. 32.5 for Centralized), demonstrating the superiority of our proposed paradigm.
>
> | Method          | En-Fr | Ar-Es | Ru-Zh | Avg  |
> |-----------------|-------|-------|-------|------|
> | Pretrained Model     | 31.4  | 27.4  | 27.9  | 28.9 |
> | PE_Centralized  | 33.9  | 31.9  | 30.7  | 32.5 |
> | PE_PT (Non-IID) (Ours)| 34.3  | 33.8  | 32.1  | 33.4 |

---

> > ### Comment · Reviewer_AG4r · 2023-11-23
> > **Comprehensive evaluation**
> >
> > Given the demonstrated performance of federated prompt tuning, I've updated the soundness score. Thank you for the followup.

---

> > > ### Author Response · Authors · 2023-11-23
> > > **Glad to Address or Discuss Any Remaining Questions or Concerns**
> > >
> > > We sincerely appreciate your feedback on our efforts to address your concerns and are very grateful for your positive support.
> > >
> > > Do you have any remaining questions or concerns about our paper that we haven't yet addressed? We are looking forward to discussing them with you on the OpenReview system, should there be any further points you wish to raise.

---

> > > > ### Author Response · Authors · 2023-11-23
> > > > **Looking forward to your responses or further suggestions/comments!**
> > > >
> > > > Dear Reviewer AG4r,
> > > >
> > > > Thanks very much for your time and valuable comments. We understand you're busy. But as the window for responding and paper revision is closing, would you mind checking our response and confirming whether you have any further questions? We are happy to provide answers and revisions to your additional questions. If our reply resolves your concern, please consider raising the overall score accordingly. Many thanks!
> > > >
> > > > Best Regards,
> > > >
> > > > The authors of Paper1909

---

### Official Review · Reviewer_LsRx · 2023-10-31

**Soundness:** 3 good
**Presentation:** 3 good
**Contribution:** 3 good
**Rating:** 5
**Confidence:** 4

**Summary:**

The paper applies federated learning on multilingual scenarios to efficiently parameter-efficient prompt fine-tuning in a manner that preserves user privacy. The idea is to utilize a single global encoder that accumulates the information via federated prompt averaging. Thus, it learns the language patterns without knowing about the user information. They evaluated the experiment on NC and XNLI datasets and found performance improvement over the baseline.

**Strengths:**

- The method is very practical since it is simple and efficient, and it is an appropriate method for training multilingual model.
- Good analysis on the data efficiency and distance measurement, showing the effectiveness of the proposed method.

**Weaknesses:**

- In terms of novelty, the proposed idea is not new, and it is only a further investigation of the multilingual setting.
- Lack of clarity. The paper does not provide enough information about how the prompts are constructed or look like and hyperparameters for all settings. I suggest adding the information to the paper or appendix.

**Questions:**

Questions:
- Do you have any findings on why multilingual centralized learning is far worse than federated learning in Table 2?
- How did you tune the training and parameter averaging?

Suggestions:
- Figure number is missing on Page 2

"As depicted in Figure , "

- Missing Figure/Table

"This translates to over 99% reduction in the communication overhead shown in 3"

- Typo

"Finetuning accuracy across different lanugages on the NC task."

---

> ### Author Response · Authors · 2023-11-20
> **Response to Reviewer LsRx [1/2]**
>
> Thanks so much for your valuable comments and feedback.
>
> ## Q1
>
> > Do you have any findings on why multilingual centralized learning is far worse than federated learning in Table 2?
>
> Yes. This phenomenon has also been observed in previous works on Federated Learning [1]. Here are some possible reasons (Section 5.1, Page 7):
>
> Firstly, **Federated Learning** has a **weight averaging** effect via the aggregation of clients’ models, which could increase the generalization of the global model, further leading to higher performance [2]. Additionally, by freezing the core language model parameters and **only learning the prompt representations**, **prompt tuning** reduces the model’s ability to overfit a dataset by memorizing specific lexical cues and spurious correlations [3].
>
> These reasons demonstrate the superiority of our federated prompt tuning over the traditional centralized finetuning paradigm, offering a more robust and generalizable approach for low-resource languages.
>
> [1] Rehman, Yasar Abbas Ur, et al. "Federated self-supervised learning for video understanding." *ECCV* 2022.
>
> [2] Izmailov, Pavel, et al. "Averaging weights leads to wider optima and better generalization."  *UAI 2018* .
>
> [3] Lester, Brian, Rami Al-Rfou, and Noah Constant. "The power of scale for parameter-efficient prompt tuning."  *EMNLP 2021* .
>
> ------
>
> ## Q2 & W2:
>
> > Lack of clarity. The paper does not provide enough information about how the prompts are constructed or look like and hyperparameters for all settings. I suggest adding the information to the paper or appendix.
>
> > How did you tune the training and parameter averaging?
>
> To further clarify the prompt tuning procedure and the hyperparameters, we've added more details in Seciton 2 and Appendix B, C in the revised version.
>
> ### Virtual Prompt
>
> We have also included a detailed figure 2 in our revised paper to more clearly show how the prompts are constructed and tuned.
>
> In general, instead of selecting discrete text prompts in a manual or automated fashion, in our Multilingual Federated Prompt Tuning paradigm, we utilize **virtual prompt embeddings that can be optimized via gradient descent**. The primary objective of each **prompt encoder **is to generate an effective prompt embedding for each client based on **task specific virtual tokens**, to guide the PLM in producing the desired outputs.
>
> Figure 2 in our revised paper shows the how our prompt tuning works on both clients and server. Specifically, a textual prompt tailored for a specific task and input text are passed to the model. Then task specific virtual tokens are retrieved based on the textual prompt. With the input text tokenized, the discrete word token embeddings are retrieved. Then virtual token embeddings are inserted among discrete token embeddings and passed together into the PLM.
>
> -----
>
> ### Federated Prompt Averaging
>
> In every communication round $t$, Federated Prompt Averaging includes the following steps.
>
> **Initialization**:
> The server initializes the global prompt encoder $h_g^{t}$. Each client initializes its local prompt encoder $h_0^{t}, h_1^{t}, \ldots h_k^{t}$.
>
> **Client Selection:**
> We select a fraction $C$ of the total $K$ clients for training. This subset size is $m = \max(C \times K, 1)$. The subset we choose is denoted as $S$.
>
> **Local Encoder Tuning:**
> Each client $k$ in $S$ fetches the current global prompt encoder $h_g^{t}$, and assembles it with the PLM. During the local training on the local data $\mathcal{D}_k$, The PLM's parameters stay fixed while local prompt encoder parameters $h_k^{t}$ are tuned.
>
>
> **Aggregation**:
> The server aggregates updates from all clients using weighted average. The global prompt encoder $h_g^{t+1}$ is updated based on the received parameters $h_k^{t}$ from clients for the next round of federated prompt tuning:
>
> $$h\_g^{t+1}=\sum\_{k=1}^K \frac{\left|\mathcal{D}\_k\right|}{\sum\_{k=1}^K\left|\mathcal{D\}_k\right|} h\_k^{t}
> $$
>
> ------
>
> ### Hyper-parameters
>
>
> For all of the experiments, we report results using the 1e-3 learning rate, and we use early stopping (5 epochs of no improvement).
>
> For FL experiments, we adjust the parameter $\alpha$ that controls the mixture of languages in the dataset. An $\alpha$ value of 1.0 signifies a uniform mixture of all languages, while values closer to 0 indicate a dominant representation of individual languages or a more separated mixture.
>
> When we use Prompt Tuning to optimize the parameter efficiency, the prompt tuning init text is *Predict the category given the following news article* for all the News Classification tasks. By providing the string of words, we initialize virtual token embeddings from existing embedding weights. This string is tokenized and tiled or truncated to match the number of virtual tokens, which is 1 in our experiments.

---

> ### Author Response · Authors · 2023-11-20
> **Response to Reviewer LsRx [2/2]**
>
> ## W1:
>
> > In terms of novelty, the proposed idea is not new, and it is only a further investigation of the multilingual setting.
>
> We would like to kindly defend our proposal.
>
> To further clarify the significance and originality of our work, we've added our motivation with Multilingual NLP Background in Appendix A and the Contribution paragraph in Section 1 in our updated paper.
>
> In the paper, we introduced federated prompt tuning as a solution to help address the linguistic and geographic boundaries hindering the application of LLMs to various regions and lower-resource languages. Here we would like to provide some clarification about the motivation and significance of our research in the following two aspects.
>
> ### Multilingual NLP and Low-resource Languages
>
> As natural language processing technologies advance, not all languages have been treated equally by developers and researchers. There are around 7,000 languages spoken in the world, and approximately 400 languages have more than 1 million speakers. However, there is scarce coverage of multilingual datasets. This is especially true for low-resource languages, where data scarcity is a major bottleneck. Furthermore, the under-indexing of certain languages is also driven by access to compute resources. Mobile data, compute, and other computational resources may often be expensive or unavailable in regions that are home to under-represented languages. Unless we address this disproportionate representation head-on, we risk perpetuating this divide and further widening the gap in language access to new technologies [1].
>
> One pressing example is biomedical data. Due to its global scale, this digital content is accessible in a variety of languages, yet most existing NLP tools remain English-centric [2].
>
> This situation highlights the need for effective strategies: how can we exploit abundant labeled data from resource-rich languages to make predictions in resource-lean languages?
>
> ### Timeliness of breaking physical and linguistic barriers in the LLM era
>
> We wanted to highlight the urgency of the problem. The problem is very timely compared to other "application scenarios." It was not even considered a year ago. Previously, due to the smaller size of language models, the demand for data was not as high, and different kinds and sources of data were treated equally. Currently, the progress of LLMs, their usability, the amount of attention they receive, and the increased regulation on data, compound and lead to the urgency of this problem, where we are among the first batch to attempt to break both lingual and physical barriers.
>
> In previous cases, data transmission was always one-directional. Existing approaches focus on solving this locally, for example, through cross-lingual transfer, as well as data augmentation and preference training to address these bottlenecks [3, 4].
>
> In our paper, we approach it from a collaborative perspective. By training LLMs collaboratively across multiple participants without sharing raw data, the accuracy, robustness, and generalizability of LLMs can be enhanced by leveraging collective knowledge and exposing models to a wider range of linguistic patterns.
>
> There exists very little research from such a collaborative perspective for low-resource languages. With data and computing power being very important yet limited for LLMs, we've never needed such a lightweight collaborative paradigm more urgently than we do right now.
>
> So we introduce the concept of "federated" as a simple and established progression to our problem, which not only contributes a timely and practical solution to a rapidly evolving field, but also vividly depicts the key innovation of our paradigm: knowledge sharing and aggregation (double direction) without data transmission.
>
> Additionally, from the federated learning perspective, as far as we know, we are the first paper to investigate the data efficiency and transferability brought by federated learning, and we believe this sheds some light on how federated learning can benefit LLM on training generalizability and stability, beyond simply mitigating compliance risks.
>
> [1] Joshi, Pratik, et al. The state and fate of linguistic diversity and inclusion in the NLP world.  *ACL 2020*.
>
> [2] Bérard et al., A Multilingual Neural Machine Translation Model for Biomedical Data *NLP-COVID19 2020*.
>
> [3] Lauscher et al., From Zero to Hero: On the Limitations of Zero-Shot Language Transfer with Multilingual Transformers, *EMNLP 2020*.
>
> [4] Xia et al., Generalized Data Augmentation for Low-Resource Translation *ACL 2019*.
>
> ---
>
> ## Suggestions:
>
> > Figure number is missing on Page 2
> > "As depicted in Figure , "
> > Missing Figure/Table
> > "This translates to over 99% reduction in the communication overhead shown in 3"
> > Typo
> > "Finetuning accuracy across different lanugages on the NC task."
> >
>
> We appreciate your detailed suggestions on the typos and missing figure/table numbers! We have fixed them all in our updated version.

---

> ### Author Response · Authors · 2023-11-23
> **Looking forward to your responses or further suggestions/comments!**
>
> Dear Reviewer LsRx,
>
> We would like to sincerely thank you again for your time in reviewing our work!
>
> We understand you might be quite busy. However, as the discussion deadline is approaching, would you mind checking our response and confirming whether you have any further concerns or questions? Any further comments and discussions are welcomed!
>
> Best Regards,
>
> The authors of Paper1909

---

### Author Response · Authors · 2023-11-23
**Summary**

We thank the reviewers for their valuable feedback and appreciate the great efforts made by all reviewers, ACs, SACs, and PCs. We are grateful that the reviewers have multiple positive impressions of our work, including: **The experiments are well-designed and the results are convincing** (xDut), **very practical since it is simple and efficient**, and **Good analysis on the data efficiency and distance measurement, showing the effectiveness of the proposed method** (LsRx), along with **Comprehensive evaluation** (AG4r).

Following the advice, we have carefully revised our draft, proofreading to correct some typos and mistakes, and completing a range of experiments to address the reviewers' concerns. Below, we provide a summary of our updates. For detailed responses, please refer to the point-by-point feedback on each comment/question and the new empirical evaluations.

* We have carefully refined the introduction to clarify the motivation and the proper claim of the contribution, to avoid potential misunderstandings regarding novelty (see Section 1). We also added a background section to discuss some practical, motivating examples of multilingual concerns (see Appendix A).

* We implemented another widely-used parameter-efficient finetuning technique, LoRA, into our paradigm, to demonstrate its superiority and effectiveness. This is in addition to our federated prompt tuning paradigm (see Table 4 and Figure 6). We conducted machine translation tasks on the pre-trained M2M100 model to support the general-purpose and significance of our paradigm. Furthermore, extensive experiments on the low-resource languages dataset, MasakhaNEWS, further demonstrate the empirical advantage for low-resource languages (see Table 3).

* We have enriched the analysis and explanations of the Federated Prompt Averaging algorithm, providing detailed clarification of the prompt construction and specifying which parts of the models are actually tuned (see Section 3, Figure 2, and Figure 3). We also improved our notation and fixed abbreviations and typos.

The above updates in the revised draft (including the regular pages and the Appendix) are highlighted in blue in our supplementary material.

**We humbly expect you could check our responses with our updated version, and confirm whether our response has addressed your concerns. More discussions are always welcome. Please let us know if there are any further questions or suggestions that we could clarify or improve.**

We once again express our gratitude to all reviewers for their time and effort devoted to evaluating our work. We eagerly anticipate your further responses and hope for a favorable consideration of our revised manuscript.

---

### Meta-Review · Area_Chair_4yNr · 2023-12-12

**Metareview:**

The paper presents a federated learning application to multilingual prompt fine-tuning of LLMs.  A strong motivation is given wrt low-resource languages and compute power, as well as with privacy.

**Justification For Why Not Higher Score:**

The paper is not technically exciting enough for a higher score, even if the application is well done.

**Justification For Why Not Lower Score:**

The application is novel and the improved results (which are a somewhat bonus byproduct of the technique)
The paper is sound (despite the problematic review mentioned above).

---

### Decision · Program_Chairs · 2024-01-16

Accept (poster)